# Tourism-Led Change of the City Centre

José Fernandes [1,*], Pedro Chamusca [2], Rubén Lois [3], Helena Madureira [1], Juliano Mattos [1] and Jorge Pinto [4]

1. Geography and Spatial Planning Research Centre (CEGOT), University of Porto, 4150-564 Porto, Portugal; hmadureira@letras.up.pt (H.M.); up200704959@up.pt (J.M.)
2. Communication and Society Research Centre (CECS), University of Minho, 4710-057 Braga, Portugal; pedrochamusca@ics.uminho.pt
3. ANTE—Grupo de Investigación de Análisis Territorial, Universidad de Santiago de Compostela, 15705 Santiago de Compostela, Spain; rubencamilo.lois@usc.es
4. Geography and Spatial Planning Research Centre (CEGOT), ISCET, 4050-180 Porto, Portugal; jpinto@iscet.pt
* Correspondence: joseriof@letras.up.pt

**Abstract:** In multicentric and increasingly complex urban regions, a city centre reinvents itself. In the case of Porto, tourism was essential for its "Baixa" renaissance. A relevant increase in visitors meant also a dramatic increase in real estate prices and significant land-use change. In field interviews, retailers noticed a "new life" before COVID-19 arrived, remarking on the positive role of tourism on urban rehabilitation and the economic viability of companies, and the negative effects for residents and traditional shops, directed to the common resident. In this article, we present and discuss its main effects in this exceptional area in Portugal's second city. We also discuss tourism dependency and the challenge of sustainability in a high-density context, defending public policies oriented for a "city with tourists" that replaces the current construction of a "city of tourists".

**Keywords:** urban tourism; overtourism; gentrification; land use; sustainability; public policies





## 1. Introduction

In recent decades, mutations in the city centre have been particularly intense as low-cost flights have become popular and changing places is easier (specially for countries within the eurozone and under the Maastricht treaty), while digital platforms facilitate travel and accommodation. As a consequence, especially historical, cultural, and environmental attractive old cities appeal to visitors and all kinds of city users. This new flowing population and real estate investment are associated with urban renewal and with soaring prices, triggering functional and residential gentrification, a term coined by Glass [1] that relates to the process of transformation of marginalized and/or traditional neighbourhoods for wealthier and more sophisticated solutions, with the replacement of residents and activities [2,3].

It is common that a process of beautification occurs, as the image of the city becomes more relevant, more so in the central squares and the façades of the most visited streets. As a consequence of social change and the process of rising prices, economic land uses also suffer significative alterations.

The main objective of this article is to discuss the change of the city centre in the face of dependence from tourism, considering "sustainable tourism" and "sustainable urbanism" principles. For that, we use Porto as a case study. And we do that for good reasons. The second city of Portugal was elected European Best Destination in 2012, 2014 and 2017, and the European Best City-Break Destination in 2020. The number of overnight stays went from 2,102,481 to 4,819,168 from 2013 to 2022 (129.2%) whereas in Lisbon the change was 78.9%, and there were 6170 Airbnb units per 100.000 inhabitants prior to COVID, in 2018, which is higher than Paris and Barcelona [4]. Also, it was in Porto that, in July 2021, 17 European cities (including Venice and Prague) and some companies (including Airbnb)

signed a document declaring their concern with excessive tourism and, in the words of the local authorities, decided to "align tourism with the best practices of sustainable policies".

To do so, our research seeks to shed light on three research questions:

- How is tourism conditioning the city centre's social and economic evolution?
- How are tourism and public policies perceived by economic agents?
- How should urbanism for sustainability be designed and implemented in a tourism-dependent city centre?

The analysis follows the following structure: In Section 2, we deal with the theoretical framework of the research. To do so, we consider the main dynamics associated with tourism in city centres and recent evolution of urban public policies in selected European city centres. Section 3 presents the materials and methods, as well as the city of Porto. Section 4 present the results for Porto city centre, and in Section 5, there is a discussion of the effects and dynamics of tourism on its relation with a sustainable approach. Finally, Section 6 presents the main conclusions and their relation to the paper's theoretical framework and research questions.

## 2. City Centre, Sustainability, and Tourism

### 2.1. Centre and Centrality: Recent Transformations

A city may have a centre or several centres, depending on its history and dimension, and the combination of the conditions for centrality (land value, symbolic value, and accessibility condition) is different in each place. Central areas differ, from the old, traditional city centre, to the new areas of the European city with high buildings which host multinationals and large companies, or new sexy and gentrified areas where culture and recreation is relevant [5,6].

Undoubtedly, the symbolic dimension of the city centre is essential and allows it to be set apart from other locations, as well as to distinguish the central city from the expanded, polycentric, and fragmented urban area. This is where major public actors (government offices, palaces, and historic universities) and large private companies (banks, insurance companies, specialized occupations offices, and luxury trade) find their place. All these activities value the city's central heritage and the highlighted examples of civil architecture of the last two or three hundred years. But heritage also bears historical significance conformed by monuments and other older buildings. That is concentrated in another type of centre, the so-called "historical" centre, since the condition of centrality is normally related here with a long period of time and the capacity the city had to retain essential elements of its construction and reconstruction over the centuries [7,8].

Historical centres and economical centres are the most attractive areas for urban tourism. The massification of access to air travel, and more economic capacity and time also means a higher value of aesthetic and cultural experiences. As a consequence, city-breaks compete with locations of 20th-century massification (sun and beach destinations), and extend far beyond the usual urban venues of Paris, London, Venice, Rome and Barcelona. As a result, in several cities, and especially in their centre, accommodation, restaurants and other tourism-oriented facilities increase dramatically.

In addition, with the emergence of the so-called "collaborative economy", best known as the "platform economy" [9,10], Airbnb and others have encouraged the multiplication of apartments and flats for rent [11,12]. These short-term rentals have helped to occupy several of the residential places that still existed in the centre of cities such as Barcelona, Lisbon, Porto, or Palma de Mallorca [13–16] and have denied about half to three quarters of the offers of new housing in New York [17]. If tourism and short-rent locations created the opportunity to invest in old housing, improving living conditions in decadent buildings, they also gave new meaning to competition in the centre regarding land uses, as it became a place where the presence of the floating population is more important than the "common population", and the price for housing, products, meals, and services become impossible for a good part of people to live and retailers to resist.

It is within this context that the rent gap applies, as the difference between the rent realized from a plot of land and the potential rent if it were developed to its "highest and best" use increases and attracts real estate interest, including international. Simultaneously, overtourism emerges as the tourist densification of specific streets and squares, occurs, along with the proliferation of hotel concentrations, forcing the displacement of the more fragile residents and activities [18,19].

*2.2. Sustainable Urban Tourism*

The relation between sustainability and tourism is based on several processes and debates, including the questions posed by the "limits to growth" concept [20,21], or the increasing perception of the negative outcomes of tourism growth in destination areas [22]. Definitions of sustainable tourism normally embody a holistic perspective, incorporating a suitable balance between economic, sociocultural and environmental aspects in long-term development perspectives [23].

Paradoxically, although the debate on sustainable tourism has been widely embraced by policy makers from both public and private organisations (although, in many occasions, some measures have been denounced as mere "green washing" exercises, by masquerading as ecological discourse for their own benefit in reducing operating costs or to keep in line with the ideological change), little to no attention has been given to the conceptualization of sustainability in an urban tourism context [24–26]. And that occurs despite the remarkable growth of urban tourism during the last two decades, the knowledge of its impacts, and the growing number of global, national, and local initiatives taken during the last decades to make cities more sustainable. Several factors may be pointed out to explain this, including the traditional focus of sustainable tourism studies on rural and eco-touristic places or the insufficient research conducted in various aspects of the urban tourism phenomenon itself [25].

Nevertheless, a growing focus on the impacts of what is seen as overtourism has led researchers to engage with sustainable urban tourism issues. The social, economic and environmental impacts of the touristification phenomena have also been increasingly acknowledged by researchers, policy makers, tourism stakeholders and local communities [18,27]. In fact, even though tourism can represent an important opportunity for development, contributing to the creation of employment and improvements in infrastructure, it can also promote the emergence or intensification of urban problems such as those related to the increase in the cost of living, new forms of gentrification and $CO_2$ emissions, as well as other forms of pollution, some of them related with congestion in circulation.

There is a challenge in achieving the best compromise between the socioeconomical benefits and the socioenvironmental negative impacts of tourism [28]. And despite the lack of a clear definition of the concept of sustainable urban tourism or the ambiguous use of the concepts of overtourism or gentrification (there are no instruments or indexes that will unequivocally define them, nor a correct geographical dimension), there is some convergence on the challenges that sustainable tourism is facing in urban areas, namely the management of conflicts between residents' quality of life and urban development processes associated with the tourism industry, or between the residents' perceived local environmental qualities and the local environmental issues induced by overtourism [24,29].

Many researchers have looked for limits and thresholds as a way to achieve sustainable urban tourism. Based on the existing studies at the time, Saarinen [22] systematised approaches to the limits of tourism growth in three main groups: (i) resource-based limits, related to the carrying capacity model and the search for a limit which cannot be overstepped without serious negative impacts on the available resources; (ii) activity-based limits, related to tourism-centric approaches and the idea that different tourism activities or segments may have different kinds of growth limits; and (iii) community-based limits, aiming to empower specifically the host communities in tourism development. However, there is not a "magic number" [22,30] for the maximum acceptable number of tourists at a destination, a threshold beyond which damage would be created, namely due to the difficulty in evaluating all former dimensions simultaneously. For instance,

carrying capacity is not only related to a certain resource but also to human values and perceptions concerning that resource [22] and it is influenced not only by the tourist's behaviours and practices, but also by the environmental and socioeconomical resilience of the destination [30].

The difficulty in defining and applying limit thresholds to sustainable urban tourism development is, in fact, a clear example of the complex nature of the concept of sustainable urban tourism and it mirrors the consequent challenges concerning its application in urban policies, for instance, the challenge of balancing the double-edged nature of tourism [31], as an important economic resource and a generator of negative socioenvironmental impacts. Or the challenge of embracing the multidimensional character of tourism, comprising complex interactions between the industry and the specific urban context.

Even though holistic sustainable urban tourism approaches should encompass both issues inherently related with the source (tourism industry, visitors' behaviours and practices) and issues related with the supply's context-specific characteristics, urban policies are almost limited to intervene, in a more defensive or reactive way, in the urban supply context.

The criticism of the conceptualization and operationalization of sustainable tourism, and particularly of sustainable urban tourism, does not diminish the importance of policies oriented by sustainable urban tourism principles. As it has been highlighted [31], despite the inherent limitations, urban policies may contribute to reduce socioenvironmental problems and to find the best compromises, minimising the negative impacts.

Sustainability has long been a part of urbanism, in different periods in history, much before tourism was important in so many cities [32]. Sustainable urbanism revolves around several pivotal principles aimed at harmonizing the interplay between environmental, socio-cultural, and economic facets in urban contexts. At its core lies the concept of compact and connected cities, emphasizing the creation of walkable neighbourhoods and mixed-use spaces to minimize urban sprawl and foster efficient land utilization as well as proximity, theoretically epitomized on the "15-min city" [33,34], in the sequence of the neighbourhood unit of Clarence Perry and the pedestrian pocket of Peter Calthorpe (see [35]). This model not only reduces reliance on cars but also encourages a sense of community and accessibility to amenities, catering to both residents and tourists.

Another crucial aspect involves prioritizing green infrastructure and biodiversity conservation. Incorporating green spaces and preserving natural habitats within cities bolsters ecological resilience, enhances air quality, and offers recreational havens for urban dwellers and visitors alike [36]. Moreover, sustainable urbanism emphasizes resource efficiency and sustainable mobility by advocating for public transportation, cycling lanes, and pedestrian-friendly pathways. This strategy aims to curb carbon emissions, alleviate traffic congestion, and promote healthier, more sustainable modes of transport [37,38].

Additionally, social inclusivity and community engagement form integral components. Ensuring access to essential services, affordable housing, and public spaces for all residents fosters a more inclusive urban environment. Community involvement in decision-making processes not only amplifies diverse perspectives but also instils a sense of ownership and belonging. These principles, along with resilience, economic prosperity, and innovation, serve as guiding pillars in urban planning and policymaking. By embracing these principles, cities can navigate the complexities of sustainable urban tourism, forging a path toward a more resilient, inclusive, and environmentally conscious urban future where tourism complements rather than compromises the well-being of both locals and visitors.

### 2.3. Policies for the City Centre

In the 1970s and 1980s, many European cities were facing physical decay, and losing both population and economic vitality. In several cases we witnessed a revival of cities and its centres as strategic places for a wide range of projects and dynamics, addressing the economic, architectural, social, cultural, and political dimensions [39–41].

The evolution of urban policies in Europe are for a large part inherited and the result of an ad hoc combination [42]. This is supported by a long-standing autonomy of cities and

municipal government, on the one hand, and the strength of states and public policies, on the other. A third aspect has been very relevant in European Union cities more recently, especially those with less investment capacity like Portugal: the impact of the EU funding programmes dedicated to urban areas and policies.

Among the multiple direct ways EU action affects urban policies, it is possible to identify programmes such as RECITE, URBAN, URBACT or INTERREG, financial engineering mechanisms such as JESSICA, research lines in urban themes under Horizon 2020, and initiatives such as the European green capital award, the European mobility week award, the smart cities stakeholder platform, Urban Audit or ICLEI Europe, Agenda 21 and the Covenant of Mayors.

The European urban policy has gone through a winding road of consolidation and maturation [43,44]. Overall, this development reproduces a framework marked by four major challenges: globalization and economic restructuring, resulting in the need to promote a more balanced urban system, with enhanced economic growth and employment; social inclusion and economic restructuring; sustainability; and governance, pointing to the increased capacity of local actors to manage change and provide better response to fiscal, organizational, institutional and administrative problems in a multi-agent and multi-scalar approach, open to public participation.

Within the context of this evolution, the urban sustainability of cities is receiving particular attention since the formalisation of the Cohesion Policy after the Single European Act, as a capital challenge; thus, it is assuming its own characteristic both as a thematic agenda of public policy and as a practical methodology of its territorialization within sustainable urban development.

## 3. Geographical Context and Methodology

### 3.1. Baixa

Porto, today, is the main centre of a polycentric urban region which, between the cities of Viana do Castelo and Aveiro, in a 150 km long and 20 km wide stretch, accounts for 3.5 million inhabitants (1/3 of the Portuguese population).

Porto is also the main administrative city of a de facto city made up of several municipalities that reaches almost one million inhabitants in a 10 km radius circle. Therefore, when we talk about the municipality of Porto, and specifically about its Baixa, we are referring to the symbolic centre and the greater geographic concentration of tourism in a vast region.

Being a place of relevant monumental density and the subject of public policies with greater prominence, Porto also plays the role of an institutional centre, having the headquarters of classical organizations such as the chamber of commerce, the regional coordination commission, various regional directorates, and the largest university in the region. In addition, Porto has an airport (Sá Carneiro) that was considered by the Airport Council International to be the best in Europe in 2022 on its category (10–25 million passengers/year), and is the great hub of the hotel industry in the region, where all the information and tourist routes of the Portuguese North are focused.

Porto has greatly increased its expression as a tourist destination. At its symbolic centre, this increase may be considered dramatic, since, 20 years ago, it survived from the effects of intense suburbanization of residence, industry and shopping. At the turn of the century, there were strong investments in the public space of the "old" city centre, in preparation for the celebration of the European Capital of Culture, in 2001. A remarkable improvement in accessibility follows, with the creation of the light rail system, where a line connecting Porto with Gaia (2005), reinforces enormously accessibility in the heart of the city, as it crosses with the east–west line at Trindade station. Low-cost aviation in a city whose historical centre was classified as a World Heritage Site in 1996 has also got to be considered as relevant in the transformation of a decadent centre. The figures of the variation between 2010 and 2019 speak for themselves: number of inbound passengers at the airport +150%; tourist accommodation +293%; traditional accommodation capacity +114%; and overnight stays +171%.

*3.2. Methods*

In this study, we combine several information sources and procedures. We use bibliography for contextualization, statistical data and other objective references, namely on public policy instruments, and the results of an intensive work of direct collection of quantitative information (with closed responses) and semi-structured interviews of a qualitative nature. We also used data resulting from previous research, such as analyses of the morphology and structure of Porto and on the Airbnb platform and short-term accommodation in the city of Porto and its recent changes [45], updated and now deepened in direct contact, namely retailers and other actors who are simultaneously residents and users of the city centre.

A functional survey conducted every two years, signalling all the changes occurring on the ground floor, was of great use in the perception of recent transformations, with the first and the most recent survey, carried out in 2012 and 2020, respectively, being considered.

An interview was applied in order to collect the local actors' opinion. The interview was held by the authors with the owners or managers of 54 establishments (Figure 1) in November 2021, chosen with attention to economic diversity, geographical distribution, and more tourist-oriented establishments and others, seemingly more oriented to the resident.

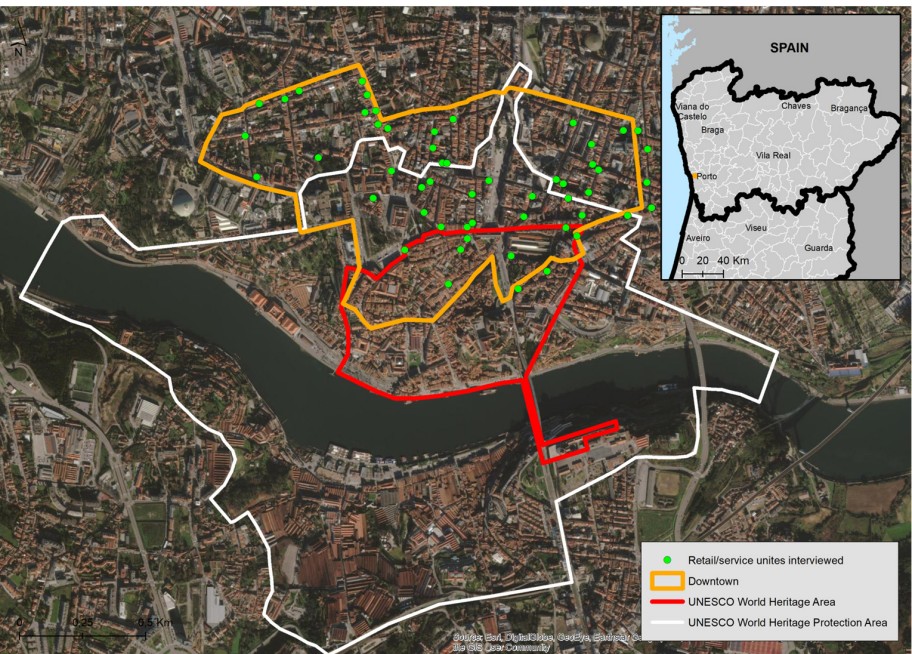

**Figure 1.** Localization of retail/service units interviewed. Source: Own elaboration.

We considered Baixa as the space already defined as the city of Porto centre (or traditional centre) in other works, keeping in mind the symbolic dimension and accessibility conditions, but, in particular, the concentration of shops and restaurants and the diversity of economic activities.

Regarding the characteristics of the interviewees, among the establishments visited, 44.4% had more than 30 years of existence and 31.5% less than 10 years. The remaining establishments fell within the range of 10 to 30 years of existence. The predominant standing was clearly the average, with only four being considered "luxury" and nine of low standing.

Regarding the diversity of activities, we visited both retail shops and restaurants and other similar units, as well as very few services of a commercial nature. The interviews included 11 shops associated with the sale of articles of personal use, 3 of alimentation articles and 17 "Horeca" units (hotel–restaurant–cafeteria).

With the answers obtained, we designed an initial framework for the characterization of change as perceived by the respondents, of the effectiveness of public policies or, on the contrary, of how processes due to strictly private dynamics were central to triggering the transformation of Baixa.

Semi-structured interviews were used in depth, following the methodological models contrasted by the bibliography. In these interviews, we wanted to know the perceived importance of tourism; the best and worst effects of recent change, and, based on their responses, to draw a predictive scenario for 2030. Finally, interviewees were asked to indicate three projects or measures to be implemented in Baixa. The semi-structured interview seeks to understand the values, opinions, behaviours, and perceptions of the respondents, based on the idea of the "city of citizens".

## 4. Results

Tourists have been essential for the transformation of land use in Porto city centre, together with other members of an increasing floating population ("digital nomad", Erasmus students, congresspersons...). Between July 2012 and July 2020, there was a strong increase in the number of tourist accommodation units (238%), coffee shops and restaurants (46.3%) and non-specialized commerce (3.9%), with a number of souvenir shops and "typical items" (see Figure 2). In 2020, the accommodation establishments (identifiable from the street), the coffee shops and restaurants accounted for 29.7% of the entire offer in Porto's Baixa, while, in 2012, they represented 17.4%.

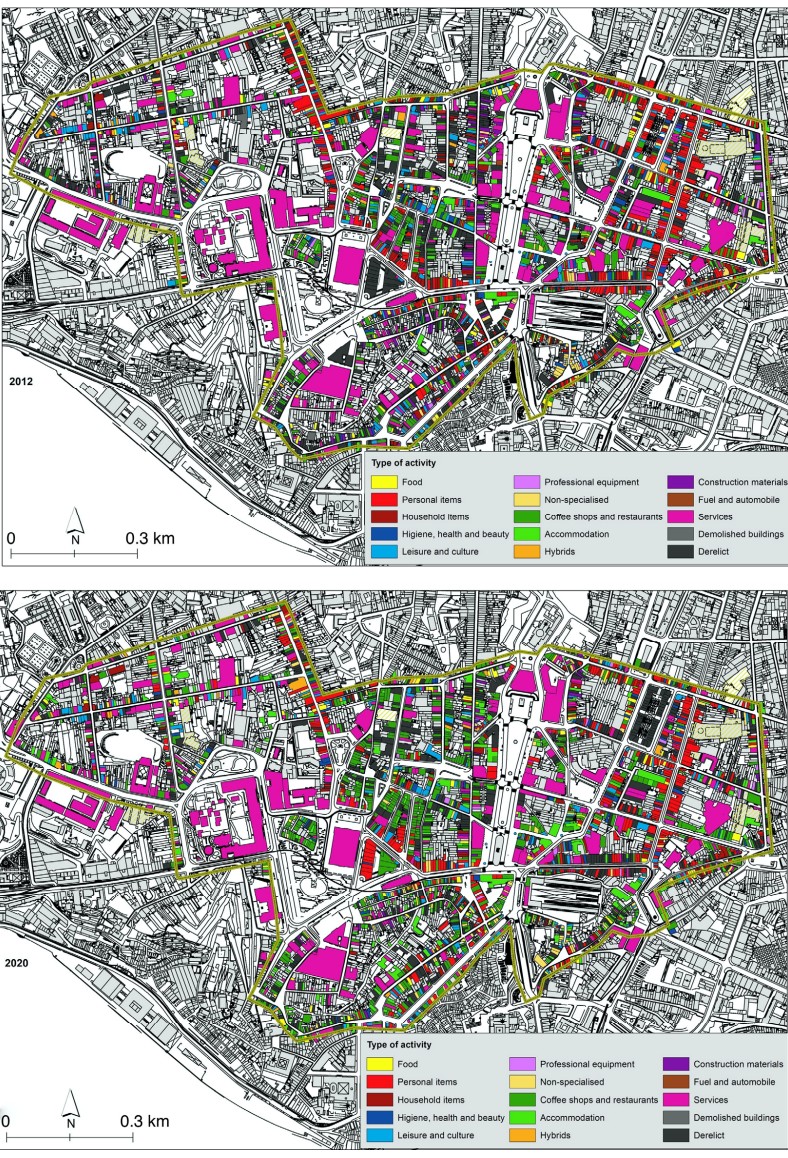

**Figure 2.** Economic activities (2012–2020) in Porto's city centre. Source: own elaboration, based on functional surveys.

In the interviews, when asked to indicate three options (out of 13) as priorities for urban policies (Table 1), it was found that economic development and housing collected 29 and 26 preferences, respectively, followed by "education, health and culture" and "accessibility and mobility" (both with 21).

**Table 1.** Porto downtown economic stakeholders' political priorities.

| Political Priority | No. of Replies |
|---|---|
| Housing | 29 |
| Economic development | 26 |
| Accessibility and mobility | 21 |
| Education, health, and culture | 21 |
| Quality of life of the populations | 18 |
| Efficient and quality governance | 13 |
| Urban planning | 9 |
| Social support | 8 |
| Engagement of citizens | 5 |
| Demographic dynamics | 4 |
| Environmental sustainability | 4 |
| Justice | 4 |
| Territorial cohesion | 0 |

This assessment was confirmed in the following answer, in which the respondent was asked to assess, from "much worse" to "much better", ten dimensions: safety, architecture, economy, housing, identity, sustainability, cleanliness, entertainment, neighbourhood and air quality (Figure 3). Here, housing stood out as the only domain that gathered the large majority of negative responses (39, with 23 of "much worse"). In all the others, except in relation to noise and safety, an improvement was noted, specifically in architecture (32 positive, 5 negative and 17 intermediate assessments), the economy (30 positive v 12 negative +12) and entertainment (27 positive v 12 negative +15).

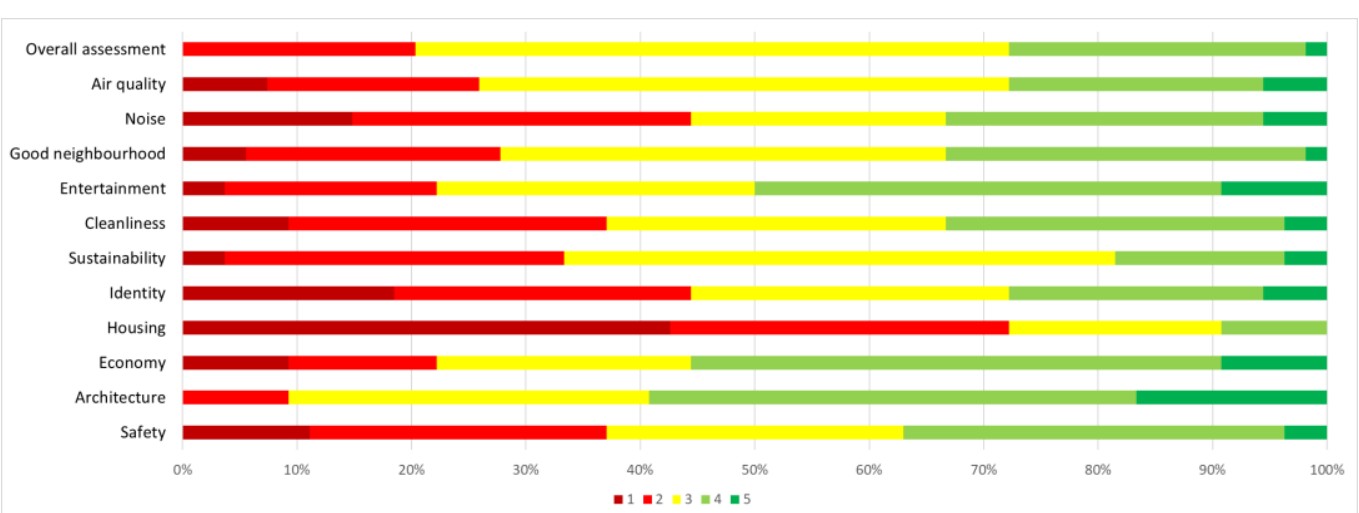

**Figure 3.** Economic stakeholders' assessment of the transformation at Porto's city centre. Source: own elaboration, based on interviews.

In the explanation for the transformations experienced in Porto's Baixa in the last decade (Figure 4), the role of low-cost flights (47 consider them important or very important), hotels (41), local accommodation (41) and real estate business (34) can be highlighted. Not so many references were made to urban rehabilitation (26) and the role of municipal policies (25). The lowest positive values were regarding housing policies (7) and social cohesion policies (12).

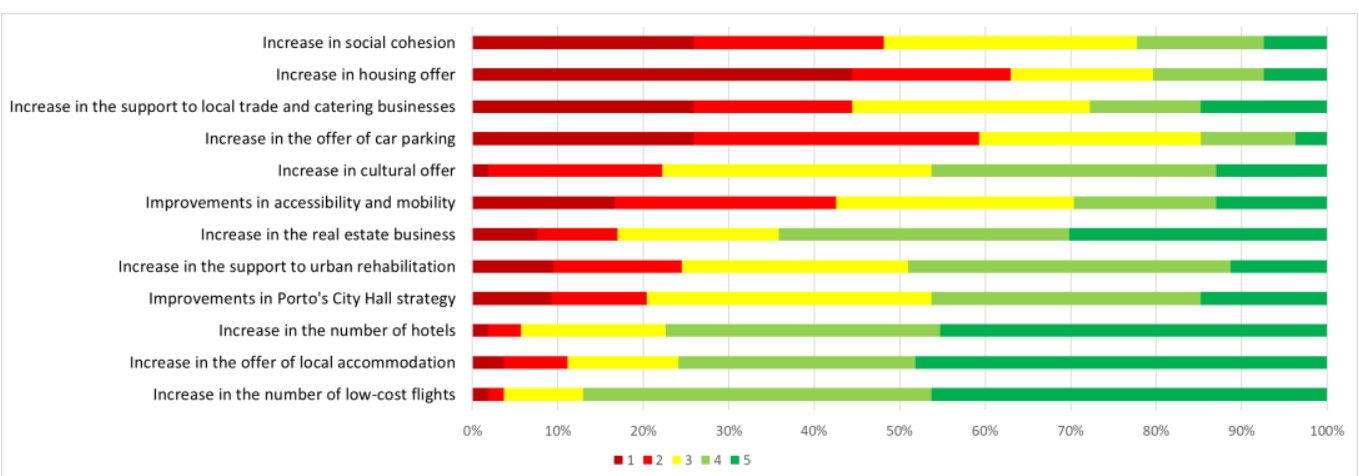

**Figure 4.** Economic stakeholders' assessment of the importance of the changes at Porto's city-centre. Source: Own elaboration, based on interviews.

During the interviews, the idea that there was a very important transformation and that tourism was at the centre of such change was universal. Tourism was referred to as "crucial", "essential", "vital", or "fundamental". But several of the respondents highlighting the positive effects on the economy and urban rehabilitation considered "expulsion" and "high rents." to be negative. Some critical voices were heard in a wider sense: "good for hotels and restaurants only", "embellished the city, but emptied it of residents", with several references to "de-characterisation" and "loss of identity".

When asked how Baixa should be, the most common references mentioned more residents, dynamism, safety, cleanliness, accessibility, justice, and green spaces.

On public policies, there was some embarrassment and contradictory ideas. Mentions included concerns with the homeless, the promotion of tourism and the struggle against gentrification, the need to balance tourism with the retention of residents and the importance of the existence of establishments capable of maintaining the identity of the city. "We need to have restaurants with Porto's traditional tripe stew", one of the interviewees said, because "the city cannot be like the 'malhão' (popular song), everyone "eating and drinking, walking in the street. . ." in the midst of nice facades (Figure 5).

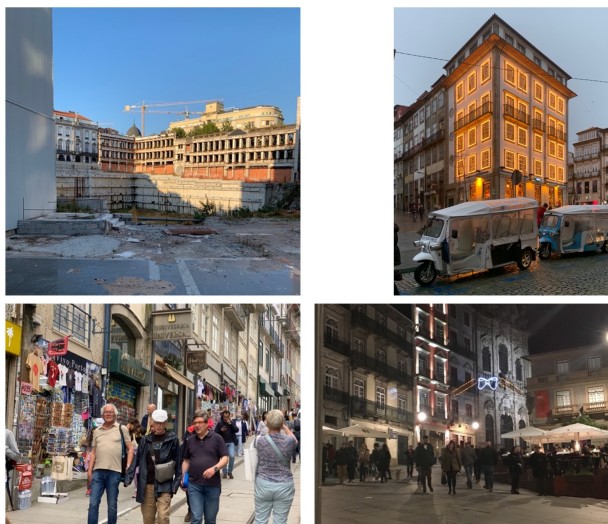

**Figure 5.** (**Left** to **right** and **top** to **bottom**)—an old façade waiting for new buildings in the back (at Sá da Bandeira and Formosa streets); a tuc tuc and a lighted building with a shop that sells canned sardines on the ground floor; souvenir shops at Rua da Assunção; São Domingos in the evening, one of many places where restaurants, museums and other "attractions" bring tourists. (Own photos).

Finally, we sought to obtain indications regarding concrete proposals. Here, the most mentioned were the measures for more and cheaper housing (5), the improvement in collective transportation (5), the limitation of hotels and local accommodation (4), the creation of more cultural events and initiatives (4), the restriction of access to cars (4), more parking (3), further cleanliness (3), support for retail (3), video surveillance (3), free parking (2), end of evictions (2), improvement of public spaces (2), limitation of restaurants (1), social support for toxic dependents (1), increased parking for residents (1), more police in the street (1), gardens (1), spaces for senior citizens (1), day-care centres (1) and pedestrian streets (1).

## 5. Discussion

Tourism was regarded as the industry of peace. This was how it was promoted in the late 1940s in Europe, when it was perceived as a generator of tolerance and an agent in the restructuring of war-torn economies, prescribed to both developed and developing countries. However, the first signs of objection and prudence related to this reductive, almost magical, vision of tourism quickly emerged. After "Tourism: Blessing or Blight?" [46], the discussion has been kept alive to the present day regarding the two sides of the coin of tourism, while incorporating new concepts and clothing, which, in a way, killed its exoticism [47].

Enjoying tourism means moving away from everyday life. The massification of tourism, the new collaborative and deregulated types of the activity, and the behaviour, often gregarious, pose a threat to the sustainability of some of the most sought-after areas or even to an entire city. Even sustainable tourism is much talked, tourism normally has an important environmental impact [11,18] and its seasonality, vulnerability to crises, and dependence from travel causes a set of perverse effects.

Considering the discussions on the limits of tourism—how much is too much?—, the emergence of tourismphobia and tourism-led gentrification [48], a new model gained general acceptability, sustainable urban development, with social and spatial justice [49], sustainability [50,51] and economic innovation [52] as the main pillars. At the same time, the EU started to support cities in tackling urban decline through explicit initiatives and programmes, thus shaping a new area-based approach, characterised by strong coordination of action, horizontal partnerships, and the concentration of funding in specific vulnerable target areas.

In Porto, after decades of house rent freezing and the incapacity of public and private sectors to promote the rehabilitation of private buildings, as well as to maintain the attractiveness of the city centre, a liberal context (both in the central and local governments, after 2002) and especially the tourism explosion of the last decade promoted a radical change, as the statistics of land use evolution and interviews demonstrate. That also brought a significant alteration in the image and character of the city resulting from urbanistic and architectonic interventions, with an emerging new concern being associated with the excessive role of tourism in the future of the city—and especially in Baixa—and its relation with the principles of sustainability. In all the cases in which hotels or apartments for short rentals pretend to instal in vacant buildings, they rarely find any resistance. When there is previous use, there is a negotiated compromise or residents with older contracts and more than 65 can stay. But there are several cases of forced eviction denounced by grassroot movements, newspapers and posters on some buildings, with no success. Tourism is to reign in (neo)liberal cities. The resulting massification of tourism in Baixa is perceived in the interviews, where the passive role of public authorities (e.g., short-term rental is free, all year, everywhere, with no limits to the number of flats you rent, their prices, or the number of days you rent them, no matter whom) is also recognized. Tourism becomes not only central but almost the only "raison d'être" of the central city, its economy, animation and urbanism. As a consequence, the strong changes in progress in the centre of the city indicate a process of touristification, and no concern is expressed by the authorities about the negative effects of these transformations nor the potential of public policies to promote more sustainable tourism.

One of the main results is the intensification of the touristification of Porto's Baixa, thus corroborating the results of previous studies, which have documented, in many ways, the transformations taking place in the city centre [3,4,45,53–56].

What may be the role of Baixa in a more sustainable city with more sustainable tourism? We have seen that the last decades have been torn by successive crises and how a territorial specialisation in tourism can be a severe fragility, especially for a city centre [57,58].

However, despite the evidence of the strong transformations in progress and signs of fragility regarding the future of the city centre, it was possible to verify the existence of divergent opinions by the economic agents interviewed regarding the impacts of these transformations. This is indicated, from the outset, by the recognition of the key role of tourism in the transformations of Porto's Baixa in the last decade and by the multifaceted assessment of these transformations, for example, observable in the contrast between, on the one hand, the positive assessment in the economy, entertainment and rehabilitation of the buildings and, on the other hand, the negative consequence in housing. These results are in line with the already-recognised "double-edged nature" of tourism [25,31].

In fact, tourism has a multifaceted nature, and it is hard to individualise and manage the complex interactions and conflicting interests. Furthermore, in the specific case of Baixa, after a long crisis in the city centre, it is easier to understand that a valuation of the positive aspects of tourism prevails, which brings us to the idea that its acceptance or challenge will not depend only on scale and its proportions, but also on the social, economic, political, and institutional context [59]. In fact, in Porto, tourism has had a pivotal effect on the disruption of a long cycle of devaluation of the city centre, which is particularly evident in the continuous decline in population, urban de-qualification and the strong expression of derelict buildings. The memory of the economic crisis of the years 2007 and 2008, which was long and intense and impacted the independent shops in the city centre in a very particular manner, must be also important in the assessment. On the other hand, and similar to what occurred in many other cities, the increase in prices and the replacement of housing units by hotels or short-term accommodation, and their effect on the expulsion of residents and the increase in the difficulty of settlement, has led many to negatively assess changes and to identify this domain as a political priority in the city of Porto.

Finally, it is important to consider the interviews in relation to public policies. The interviewees' perception that tourism simultaneously stimulates positive and negative impacts on the city and on the quality of life of its inhabitants will certainly be one of the explanatory factors for the difficulties felt in envisioning the potential of public policies in the construction of a better city centre. On the other hand, the answers recognise the limited role of public policies in the management of tourism in the city, which is well evidenced in the residual weight attributed to policy measures (e.g., municipal policy, housing policy, and urban rehabilitation policy) in explaining the transformations experienced in the last decade. The results may perhaps be corroborating the acknowledged difficulty of devising public policies capable of responding to a complex system [25,31], which involves multiple agents from different fields and scales, and in which it is difficult to untangle the web between tourism, the city and the well-being of its residents.

But to address the challenges of overtourism in Porto while fostering sustainable urbanism, a multifaceted approach is possible, and seems essential. Firstly, zoning regulations that limit short-term rental, mitigating residence prices, evictions, and gentrification should be put in place. Establishing a balance between tourist accommodations and local housing is also crucial to achieve a long-standing diverse and sustainable community. Additionally, the necessity for new housing or rehabilitation to be mostly mixed-use, combining residential spaces with cultural and commercial areas, could help to reinforce the complexity of the city.

To preserve Porto's heritage and support local businesses, policies favouring historical shops and traditional retail in general as well as local artisans and regional products over generic tourist-oriented establishments and goods can help maintain the city's character.

In the same direction, work on buildings should transform the minimum and involve the least amount of production of construction materials as possible, as well as travel for goods and workers.

Strategic urban planning with a sustainable perspective should prioritize public spaces and pedestrian-friendly zones, and a reduction in the environmental impact of circulation. Investing in efficient public transportation systems, cycling infrastructure, and walkable areas not only eases congestion but also promotes eco-friendly travel.

Furthermore, fostering community engagement through participatory decision-making processes empowers residents, and their ideas will generally contribute to more sustainable urban development. The collaboration of local organizations, residents, and businesses in eco-tourism initiatives, such as special guided tours and community-led cultural events, are a good example of how local participation can enhance visitor experiences while minimizing environmental impact.

In short, a comprehensive strategy encompassing regulatory measures, thoughtful urban planning, community involvement, and responsible tourism promotion is crucial for ensuring Porto's sustainability as a tourism-dependent city, as well as intermunicipal coordination.

## 6. Conclusions and Comments

The research carried out in the city of Porto allows us to understand better the causes, characteristics and consequences of recent change in Baixa. We realised that the transformation of Porto's city centre, much oriented towards the tourist, has been promoting a process of residential and functional gentrification, as Carvalho, Chamusca, Fernandes and Pinto [54] noted. The multiplication of a new type of retail and service unit is associated with the increase in the price of the land (resulting from an increased tourist demand and urban rehabilitation processes, as well as the legal easing of the tenancy law), thus triggering the closure of many establishments and the displacement of others. The increase in tourist demand has also reinforced the real estate market attractiveness, leading owners (much of them new, and some being international funds) to significantly increase rents for housing, which, together with changes in the law, has led to evictions and a transformation of the social and economic pattern of the resident, with a significant increase in the non-permanent inhabitant. In fact, the National Institute of Statistics data show that the real estate selling price has grown 38.4% in the municipality of Porto and 44.7% in the historical centre parish between 2019 and 2022 (Portugal average was a 35.6% increase).

Tourism is identified as the main engine of transformation in the city centre. The vast majority of respondents highlight its effects on the economy and urban rehabilitation. However, they also notice that this urban revitalization came at a very high cost, particularly social, including the departure of many residents and small entrepreneurs, due to the widespread increase in land cost and the general cost of living.

Finally, the need for an integrated approach is evident from the interviews. The COVID-19 pandemic emphasised the city's enormous dependence on tourism and the unsustainability of this model of high specialization. As a consequence, Porto is a good reference for reflection on the post-COVID relationship between the city and tourism. The conclusion, supported by the interviews, allows us to argue for the advantage of a demand geared towards visits which are longer in time and calmer in speed. And, on the city side, for the need of an urban policy less eager to capture revenue and promote infinite inflows. These guidelines could be the foundation of a policy that on the one hand, prevents the mono-functionality of a "city to sleep, eat, drink and photograph" [60] (p. 53), and on the other hand, avoids the continuous increase in the number and distance of commutes, resulting from the residents' need to seek affordable residence, with the low salaries they earn in the establishments that tourism creates, and, finally, responds to the need for more sustainable tourism and a sustainable city, responding to the increase in avoidances of tourists from the most famous attractions [61].

More than a policy for tourism or a policy for the central city, the city of tourist consumption, in the face of the paradigm of sustainability, forces us to consider control over areas of overtourism and a multiscale and integrated perspective. A sound strategy will only be achieved if it is designed to articulate the centre with the entire urban area, and a strategy for tourism, in any territory and for the most part in a city, must be part of a more comprehensive strategy that considers interests other than just economic or short-term.

**Author Contributions:** Conceptualization, J.F., H.M., J.P., P.C. and R.L.; methodology, J.F., J.P., J.M. and P.C.; investigation, J.F., J.P., J.M. and P.C.; writing—original draft preparation, H.M.; writing—review and editing, J.F.; visualization, P.C.; supervision, J.F.; project administration, J.F. All authors have read and agreed to the published version of the manuscript.

**Funding:** This work received support from the Centre of Studies in Geography and Spatial Planning (CEGOT), funded by national funds through the Foundation for Science and Technology (FCT) under the reference UIDB/04084/2020, as well as from FCT financed projects UIDB/04084/2020, UIBD/00736/2020, and PTDC/GES-URB/30551/2017.

**Data Availability Statement:** Data is contained within the article.

**Conflicts of Interest:** The authors declare no conflicts of interest.

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
