# Peer review of "Tourism-Led Change of the City Centre"

_land, doi:10.3390/land13010100_

Round 1

Reviewer 1 Report

Comments and Suggestions for Authors

The research question is coherent with actual sustainability’s issues and is well defined.

The results provide an interesting starting point of a reflection about the impacts of flowing population and tourists in particular in the transformation of the land use of Porto city centre, linking them also to sustainability aspects.

The paper presents a clear and specific methodology to demonstrate the thesis.

The article is written using a simple language and the use of the English language is very appropriate and understandable.

The issues addressed in the paper are of great interest to readers from different scientific fields and represents a very interesting starting point for other experimentations which should be applied also to other sectors. 

The paper proposes work that can be further implemented but opens interesting perspectives to operationalize sustainability goals in tourism sector.

I would pay attention to the paragraph on discussions, which does not seem coherent to me. Some parts still seem descriptive of the case study and it seems anomalous to find images in this paragraph. The discussions should focus exclusively on commenting on the results obtained with respect to the methodology adopted and contextualising them with respect to the case study without adding descriptive elements which, in my opinion, could have been included in the previous framing paragraph.

Author Response

We thank you for the appreciation of the article and his/her indications about its quality and interest to readers. According to recommendations that were made we paid special attention to point 5 “Discussion”. We accepted the suggestion to take out pictures from this part. They were included in part 4 “Results”. But we ask for comprehension to maintain some references on a general discussion that goes behind what comes out from the interview, which in our view allows a better understanding of the challenges confronted by the city centre of Porto.

Reviewer 2 Report

Comments and Suggestions for Authors

The text under review could improve its quality introducing in the section of discussion a mention to the social movements of resistance to Porto's gentrification. The social response, initially linked to the platform "Stop despejos" and to movements as "Assembleia de Moradoras e Moradores do Porto" has even manifested through artistic practices of resistance. It would be interesting to mention these reactions from civic society. Its trace (banners, logos, etc.) has been seen on the facades of many of the buildings in the city centre.

Author Response

We thank you for the suggestion. The identification of the position and the action of the parts – real estate companies / landlords / business owners / tenants / … – was not the focus of the article. But mention to conflicting interest and resistance from residents and grassroots movements was added as a welcomed improvement to the general understanding of recent change of the city centre of Porto.

Reviewer 3 Report

Comments and Suggestions for Authors

Thank you for allowing me to read this manuscript. The content is uneven with the analytical component of some interest and specific sections quite undertheorized and underdeveloped (sections 2.2., 2.3, and 5). The intended focus of the paper on tourism and sustainability is rather weak and, I suspect, it reflects the author/s lack of sensibility towards environmental and socio-economic resilience questions. The section on public policies is also extremely vague and of little use to the discussion of findings or implications for action. The methodology ought to be strengthened as well. It would seem to me that the interviews could be more accurately designated as a survey. Therefore, I cannot be optimistic about the manuscript making it to Land.

Detailed comments:

¾    The title is way longer than needed. Please shorten it.

¾    Abstract: Where is the evidence that the center resists? Sleeping rough on the front steps of stores and under viaducts? Why is tourism essential? During the pandemic only essential services were allowed to run, and to my recollection, tourism wasn’t in said classification. Tourism was also not essential when Porto was known for being an industrious city, which produced real things of value. The abstract’s last sentence ought to be re-written for clarity.

¾    Gentrification is never defined in its most understandable and assertive terms (i.e. the displacement and ruthless expulsion of existing residents to other places).

¾    Negative effects for traditional shops? Don’t tourists like to shop in traditional shops?

¾    Why is sustainability central to the future of baixa? Shouldn’t it be central to the future of the whole metropolitan area and of the country for that matter? The geographical orientation of the paper detracts from larger unsustainable processes at play in the metropolis and the country.

¾    Is it really the most appropriate way to begin a paper about Portugal’s second city with a reference to Chicago’s School of Urban Ecology? The comment about Chicago also applies to the role of air travel in the city’s economy. Said argument could perhaps make more sense after introducing what has happened and is happening on the ground. I would begin with the endogenous factors instead, those that ought to be controlled by public policies such as metropolitan sprawl development, territorial cohesion, traffic congestion, land use planning, etc.

¾    Care to give examples of other academics?

¾    What are examples of “all kinds of non-permanent residents”?

¾    Why the reference to parallel real estate? It is dubious whether this refers to the built environment or short-term rentals.

¾    Was the increase in tourism in Porto disproportional from what was experienced in, for instance, Lisbon or Barcelona? Providing comparative figures would clarify the doubt and bring greater credibility to the claim. The same applies to the AirBnB figure. How does that number fare in comparison to those of other tourism-driven cities in similar cities? So what test?

¾    “tourism-dependent” is an awkward way to think about a city center. Fortunately, city centers are (and ought to remain) about much more than tourism; think about the organic mix of functions that has traditionally characterized it.

¾    The last paragraph of the introduction ought to be more direct.

¾    Invoking Palma de Mallorca as comparable to Porto seems quite an inadequate stretch of the imagination; a mediterranean island economy doesn’t compare that well with a country’s second largest metropolitan area.

¾    There are multiple references to the rent gap theory: however, it is inadequately explained in the text.

¾    The sustainability sub-section is rather confusing and without a definition that can be used to guide the discussion. In my view, it ignores the most cutting-edge scholarly work on this topic.

¾    “maintain and improve what has already been done” – One could argue that it is because of such hasty interpretations of sustainability that some people are suffering disproportionately with the current impacts of climate change, which simultaneously preclude many to live dignifying lives.

¾    Overtourism is much broader than “tourist concentration in cities and particularly in their historical centres.” It also occurs in nature preserves, national parks, heritage areas, popular coastal areas and beaches, etc.  Many of those places have successfully implemented “destination governance” mechanisms to control the less positive impacts of too much tourism demand. The claim that there are no instruments or indexes to, at least, attempt to control tourism demand (let alone sickening speculatory supply) is untrue. While one number alone may not be the most adequate, I am reminded of the British Town Center Management (TCM) audits conducted on a regular basis by TCM managers and their frequent monitoring and  reporting of performance indicators to interested stakeholders.

¾    Please be careful when you make this sort of claim: “By the mid or the end” – imprecisions of this magnitude make one question the rigor of the analysis conducted.

¾    Section 2.4 on public policies is also extremely weak and of little use to the discussion of findings. Simply invoking the acronyms of EU funding programs does not say much about their effectiveness or impact.

¾    Fig.1 – I suggest changing the color of the “UNESCO protection area” to white, so that it has a better contrast and is more readable. The area demarkated as baixa seems too generous and not in consonance with the geographical area commonly found in other papers about Porto’s city center; I am referring to the top left corner near the city’s public library and the Crystal Palace pavilion.

¾    Fig. 2 Demolished and derelict categories are not types of activities per se, and they ought to either be removed or the figure’s title and key changed to better reflect the results.

¾    The font size of Figs. 3 and 4 is too small and unreadable. It would make more sense to add keys to the numerical scale 1–5.

¾    The results and the graphical displays seem to point to a survey and not to interviews. Either way, there are no details about when those were conducted, by whom, and whether certain business owners or managers refused to take part in the study.

¾    Cities of citizens? Aren’t they all?

¾    Opening a paragraph with the expression “Flowing population” seems quite odd and vague.

¾    Table 1 could be organized by decreasing order of replies instead.

¾    It was stated before that there is no “magic number” but then the coin metaphor is invoked to refer to two sides and that exoticism has suffered a tragic fate;  how was the “corpse” autopsied to make such a discovery?

¾    Furthermore, it is written that “tourism is anything but sustainable” – how come the Journal of Sustainable Tourism has been in print since 1993?

¾    “short term rental is free” – care to explain?

¾    “Tourists and other floating people” – who exactly are those “other people”? This tone likely is not what keeps attracting visitors to Porto or to make it the best tourist destination in Europe.

¾    The paper’s main finding seems to be that there has been an Intensification of tourism, which corroborates other studies. How many corroborations are needed to change course?

¾    The discussion ought to be a more in-depth analysis of the interviews’ (survey?) results and their articulation with the theoretical postulates introduced in the first part of the paper. As it stands, it seems way too vague and not backed by the theoretical framework in the first part of the paper.

¾    It would also be helpful to provide some evidence on the increase in real estate values.

¾    At least one reference lacks Eds.

Comments on the Quality of English Language

n/a

Author Response

We thank you very much for your in depth reading and valuable comments allowing for considerable improvement of the article.

We answer directly to your comments and ask you to please check the new version of the article. We are very glad with the improvement that resulted from your critics.

¾    The title is way longer than needed. Please shorten it.

Done.

¾    Abstract: Where is the evidence that the center resists? Sleeping rough on the front steps of stores and under viaducts? Why is tourism essential? During the pandemic only essential services were allowed to run, and to my recollection, tourism wasn’t in said classification. Tourism was also not essential when Porto was known for being an industrious city, which produced real things of value. The abstract’s last sentence ought to be re-written for clarity.

It has been changed. But it is disputable that most European city centres are “sleeping rough on the front steps of stores and under viaducts”. And surely tourism was essential in the case of Porto “old centre” recovery, as respondents to the interview notice.

¾    Gentrification is never defined in its most understandable and assertive terms (i.e. the displacement and ruthless expulsion of existing residents to other places).

      Thanks for the idea. We added a definition on lines 40-43. About displacement also see what is added on p. 3, please.

¾    Negative effects for traditional shops? Don’t tourists like to shop in traditional shops?

      We don’t mean historical shops, but old and not so attractive shops. For the idea to become clearer we completed that sentence that is now: “and negative effects for residents and traditional shops, oriented to the common resident”

¾    Why is sustainability central to the future of baixa? Shouldn’t it be central to the future of the whole metropolitan area and of the country for that matter? The geographical orientation of the paper detracts from larger unsustainable processes at play in the metropolis and the country.

      Sustainability is important to all the planet, and should be considered at all scales. In this case we are treating the city centre. Anyway, to avoid misunderstandings, we changed the sentence to “We also discuss tourism-dependency and the challenge of sustainability in a high-density context, defending public policies oriented for a “city with tourists” that replaces what seems to be a “city of tourists” vision.

¾    Is it really the most appropriate way to begin a paper about Portugal’s second city with a reference to Chicago’s School of Urban Ecology? The comment about Chicago also applies to the role of air travel in the city’s economy. Said argument could perhaps make more sense after introducing what has happened and is happening on the ground. I would begin with the endogenous factors instead, those that ought to be controlled by public policies such as metropolitan sprawl development, territorial cohesion, traffic congestion, land use planning, etc.

      The mention to the Chicago’s School of Urban Ecology intended to put in the context the interest for the study of the city centre. It was deleted. We kept our preference for a general view before getting into Porto, though.

¾    Care to give examples of other academics?

      We think that we don’t need to add examples after the first paragraph being deleted.

¾    What are examples of “all kinds of non-permanent residents”?

      It was replaced by “other city users”.

¾    Why the reference to parallel real estate? It is dubious whether this refers to the built environment or short-term rentals.

      “Parallel” was deleted. It was not the better word to mean “consequent” or “simultaneous”, and we think that a word is not necessary to join “floating population” and “real estate investment”.

¾    Was the increase in tourism in Porto disproportional from what was experienced in, for instance, Lisbon or Barcelona? Providing comparative figures would clarify the doubt and bring greater credibility to the claim. The same applies to the AirBnB figure. How does that number fare in comparison to those of other tourism-driven cities in similar cities? So what test?

      It was not disproportional: values are added for Lisbon. But it was “Best Destination” more times, has higher density of Airbnb units, and it was there that top Europe urban destinations met to discuss sustainable tourism (as it is mentioned).

¾    “tourism-dependent” is an awkward way to think about a city center. Fortunately, city centers are (and ought to remain) about much more than tourism; think about the organic mix of functions that has traditionally characterized it.

      We agree. But the fact is that, in some cities, as Porto, city centres are sadly becoming “tourism-dependent”

¾    The last paragraph of the introduction ought to be more direct.

      We changes, and  hope it is now more direct.

¾    Invoking Palma de Mallorca as comparable to Porto seems quite an inadequate stretch of the imagination; a mediterranean island economy doesn’t compare that well with a country’s second largest metropolitan area.

      On line 165 Palma da Maiorca is not compared to Porto. It is only invoked, just as one of many cases where “short-term rentals have helped to occupy several of the residential places that still existed in the centre of cities”.

¾    There are multiple references to the rent gap theory: however, it is inadequately explained in the text.

      A reference is included in p. 3.

¾    The sustainability sub-section is rather confusing and without a definition that can be used to guide the discussion. In my view, it ignores the most cutting-edge scholarly work on this topic.

      After collective reflection we decided to give up of 2.2.

¾    “maintain and improve what has already been done” – One could argue that it is because of such hasty interpretations of sustainability that some people are suffering disproportionately with the current impacts of climate change, which simultaneously preclude many to live dignifying lives.

      We could not find such an affirmation.

¾    Overtourism is much broader than “tourist concentration in cities and particularly in their historical centres.” It also occurs in nature preserves, national parks, heritage areas, popular coastal areas and beaches, etc.  Many of those places have successfully implemented “destination governance” mechanisms to control the less positive impacts of too much tourism demand. The claim that there are no instruments or indexes to, at least, attempt to control tourism demand (let alone sickening speculatory supply) is untrue. While one number alone may not be the most adequate, I am reminded of the British Town Center Management (TCM) audits conducted on a regular basis by TCM managers and their frequent monitoring and reporting of performance indicators to interested stakeholders.

      The sentence was eliminated. TCM is a good example. Rare and with no success in Portugal.

¾    Please be careful when you make this sort of claim: “By the mid or the end” – imprecisions of this magnitude make one question the rigor of the analysis conducted.

      Terrible translation from the Portuguese. Replaced by “On 70’s and 80’s of the 19th century”

¾    Section 2.4 on public policies is also extremely weak and of little use to the discussion of findings. Simply invoking the acronyms of EU funding programs does not say much about their effectiveness or impact.

      We think it is ok now.

¾    Fig.1 – I suggest changing the color of the “UNESCO protection area” to white, so that it has a better contrast and is more readable. The area demarkated as baixa seems too generous and not in consonance with the geographical area commonly found in other papers about Porto’s city center; I am referring to the top left corner near the city’s public library and the Crystal Palace pavilion.

      We think it is ok now.

¾    Fig. 2 Demolished and derelict categories are not types of activities per se, and they ought to either be removed or the figure’s title and key changed to better reflect the results.

      We think it is ok now.

¾    The font size of Figs. 3 and 4 is too small and unreadable. It would make more sense to add keys to the numerical scale 1–5.

      We think it is ok now.

¾    The results and the graphical displays seem to point to a survey and not to interviews. Either way, there are no details about when those were conducted, by whom, and whether certain business owners or managers refused to take part in the study.

      We think it is ok now.

¾    Cities of citizens? Aren’t they all?

      No, all of those that are in a city are citizens. A lot of the people living in our cities today don’t have essential rights of citizenship as the right to vote, not to mention the right to free health care or education of residence. And… are tourists (floating) residents?

¾    Opening a paragraph with the expression “Flowing population” seems quite odd and vague.

      Changed by: The tourists have been essential for the transformation of the land use of Porto city centre together with other members of an increasing floating population (“digital nomad”, Erasmus students, congresspeople, …).

¾    Table 1 could be organized by decreasing order of replies instead.

      We think it is ok now.

¾    It was stated before that there is no “magic number” but then the coin metaphor is invoked to refer to two sides and that exoticism has suffered a tragic fate; how was the “corpse” autopsied to make such a discovery?

      We think it is ok now.

¾    Furthermore, it is written that “tourism is anything but sustainable” – how come the Journal of Sustainable Tourism has been in print since 1993?

      Even there’s a nice journal with that name it does not mean that tourism is sustainable. But we agree that as it was written it a too strong declaration, after all you may do tourism walking and picking your own food. So, we change for “Even sustainable tourism is much talked, the fact is tourism normally has an important environmental impact [11,18]”

¾    “short term rental is free” – care to explain?

      We should. Thank you for noticing the need for that. We added in (e.g., short term rental is free, all year, everywhere, with no limits to the number of flats you rent, their prices, or the number of days you rent them, no matter whom).

¾    “Tourists and other floating people” – who exactly are those “other people”? This tone likely is not what keeps attracting visitors to Porto or to make it the best tourist destination in Europe.

      It was explained who they are the other, e.g. “digital nomad”, Erasmus students, congresspersons, …).

¾    The paper’s main finding seems to be that there has been an Intensification of tourism, which corroborates other studies. How many corroborations are needed to change course?

      That is a political question, not an academic one. We are very clear on the conclusions on the last two paragraphs. And stressed that even more now.

¾    The discussion ought to be a more in-depth analysis of the interviews’ (survey?) results and their articulation with the theoretical postulates introduced in the first part of the paper. As it stands, it seems way too vague and not backed by the theoretical framework in the first part of the paper.

      We think it is ok now.

¾    It would also be helpful to provide some evidence on the increase in real estate values.

      We think it is ok now.

¾    At least one reference lacks Eds.

      We think it is ok now.

Round 2

Reviewer 3 Report

Comments and Suggestions for Authors

I have reviewed the revised version of land-2782444 carefully. I am glad that the authors accepted some of my comments and attempted to improve it.

However, I am convinced that the manuscript does not do what it promises. The simple removal of the “sustainability in urban context” sub-section, together with the elimination of a number of references, instead of its reformulation with the most cutting edge knowledge on the topic as suggested in my initial review, has precluded the authors from adequately fulfilling the manuscript’s initial objective (also maintained in the current version): “to discuss change of the city centre in the face of dependence from tourism, considering “sustainable tourism” and “sustainable urbanism” principles,” as well as answering the authors’ own sub-question (also maintained in the current version): “How should urbanism for sustainability be designed and implemented in a tourism-dependent city centre?”

As stated earlier, it is my opinion that the mostly corroboration of previous studies on the topic (some conducted by the same authors) does not add enough new material, especially on tourism management and destination governance (aka: what to do to change the unsustainable tourism situation) to warrant its publication in Land. 

Comments on the Quality of English Language

Moderate editing of English language required

Author Response

Regarding the remaining criticism of referee 3 about the non answer to our question on “How should urbanism for sustainability be designed and implemented in a tourism-dependent city centre" we don't agree on that as we believe that the previous version had several ideas that could be seen as answers to the question. But we were happy to give it another look and to be more precise, also adding examples, also saying more about "tourism management and destination governance" in lines 432-464 and 780-807.